# Effects of *Artemisia ordosica* Crude Polysaccharide on Antioxidant and Immunity Response, Nutrient Digestibility, Rumen Fermentation, and Microbiota in Cashmere Goats

**DOI:** 10.3390/ani13223575

**Published:** 2023-11-19

**Authors:** Shuyi Li, Yongmei Guo, Xiaoyu Guo, Binlin Shi, Guoqiang Ma, Sumei Yan, Yanli Zhao

**Affiliations:** Inner Mongolia Key Laboratory of Animal Nutrition and Feed Science, College of Animal Science, Inner Mongolia Agricultural University, Huhhot 010018, China; lsy2835245597@163.com (S.L.); ymguo2015@163.com (Y.G.); gxy_2594@163.com (X.G.); shibinlin@yeah.net (B.S.); mgq19930120@163.com (G.M.)

**Keywords:** *Artemisia ordosica*, natural extracts, animal health, ruminal microbiota, cashmere goat

## Abstract

**Simple Summary:**

Semi-intensive and extensive captive rearing systems for cashmere goats have been gradually adopted due to the limited availability of pasture and the seasonal imbalance of nutrients in grazing pastures in China. However, intensive rearing increases cashmere goat oxidative stress risk while improving lamb productivity and economic efficiency. Crude polysaccharide from *Artemisia ordosica* has excellent hydrophilicity and various biological activities. It can alleviate oxidative stress with its antioxidant, antitumor, and immunity-enhancing effects. In this laboratory, the addition of 0.3% *Artemisia ordosica* crude polysaccharide (AOCP) improved rumen fermentation in cashmere goats, as confirmed by in vitro rumen fermentation analysis. This experiment investigated the effect of AOCP supplementation in the ration on the antioxidant and immune functions of the cashmere goat. The results of the study indicated that the addition of 0.3% AOCP to the ration improved rumen fermentation, antioxidant and immune functions, nutrient digestibility, and growth performance of cashmere goats by altering rumen flora distribution, promoting beneficial bacterial colonization, and decreasing the colonization of potentially pathogenic bacteria. Therefore, AOCP may be a viable source of antioxidants for cashmere goats.

**Abstract:**

The objective of this experiment was to investigate the effect of dietary supplementation with *Artemisia ordosica* crude polysaccharide (AOCP) on growth performance, nutrient digestibility, antioxidant and immunity capacity, rumen fermentation parameters, and the microbiota of cashmere goats. A total of 12 cashmere goats (2 years old) with similar weight (38.03 ± 2.42 kg of BW ± SD) were randomly divided into two dietary treatments with six replicates. The treatments were as follows: (1) control (CON, basal diet); and (2) AOCP treatment (AOCP, basal diet with 0.3% AOCP). Pre-feeding was conducted for 7 days, followed by an experimental period of 21 days. The results showed that the ADG; feed/gain (F/G); and the digestibility of DM, CP, and ADF of cashmere goats in the AOCP group were greater than in the CON group (*p* < 0.05). Still, there was no significant effect on the digestibility of EE, NDF, Ca, and P (*p* > 0.05). Compared to the CON group, AOCP increased BCP, propionate, butyrate, isobutyrate, valerate, isovalerate, and TVFA concentrations (*p* < 0.05), but it reduced the protozoa numbers of acetate and A/P (*p* < 0.05). The serum CAT, GSH-Px, T-SOD, 1L-6, and NO levels were higher in AOCP than in the CON group (*p* < 0.05). The addition of AOCP increased the Sobs and Ace estimators (*p* < 0.05) and reduced the Simpson estimator in the ruminal fluid compared to the CON group (*p* < 0.05). Additionally, the AOCP group increased the colonization of beneficial bacteria by positively influencing GSH-Px and IL-6 (*norank_f__F082*, *unclassified_p__Firmicutes*), as well as bacteria negatively associated with F/G (*norank_f__norank_o__Bacteroidales*, *unclassified_p__Firmicutes*, and *norank_f__F082*). It decreased the colonization of potential pathogenic bacteria (*Aeromonas* and *Escherichia-Shigella*) (*p* < 0.05) compared to the CON group. In conclusion, 0.3% AOCP improves the growth performance, nutrient digestibility, antioxidant status, immune function, rumen fermentation, and microflora of cashmere goats.

## 1. Introduction

The Inner Mongolian Arbas White cashmere goat breed has been developed through long-term natural selection, along with artificial breeding for both cashmere and meat. It is well known for its excellent pure white, thin, long, and soft cashmere. In addition, Arbas white cashmere goat meat has become more and more popular because of its rich nutrients and delicious flavor [1]. However, due to the seasonal imbalance of nutrients in pastures and the limited supply of pasture, semi-intensive and intensive yard-farming systems for cashmere goats have been gradually adopted [2]. While intensive feeding boosts lamb productivity and economic effectiveness, it also raises the risk of sub-health conditions occurring [3]. This is because it causes an imbalance between the body’s antioxidant and oxidative systems and generates large amounts of oxidative intermediates like reactive oxygen species (ROS), whose elevated levels cause poor health and metabolic diseases in cashmere goats, such as the higher prevalence of caseous lymphadenitis in intensively farmed sheep than in conventional sheep [4]. To a certain extent, antibiotics can alleviate these problems, but their disadvantages of leaving easy residues and resulting in resistant bacteria are insurmountable.

The use of phytoextracts and active botanical ingredients in animal production has received considerable interest in recent years, mainly due to increased efforts to reduce both the environmental burden of drug use and antimicrobial resistance. Among the various uses, these extracts have been proposed as feed ingredients or water supplements for their nutraceutical, probiotic, or immunomodulating effect, or for their influence on the final product quality [5]. There are abundant *Artemisia ordosica* resources in Inner Mongolia and Shaanxi in the northwest of China. It is one of the most common members of the family *Artemisia* ceae and rich in polysaccharides, flavonoids, and essential oils [6]. It has been demonstrated in numerous studies that the aqueous extract of *Artemisia ordosica* shows antioxidant and anti-inflammatory properties in animals like mice [7], broilers [8], and pigs [9], making it one of the most promising antibiotic alternatives currently available. However, no application studies in sheep have been reported. Among the effective compounds of *Artemisia ordosica* crude polysaccharide, polysaccharides have better hydrophilicity and a variety of biological activities, such as enhancing cell-mediated antioxidant [10], anti-inflammation [11] and antitumor [12] activities. Broilers stimulated by LPS (100 g/kg) exhibit reduced immune function and production performance that is alleviated by AOCP [13]. Furthermore, our previous in vitro rumen fermentation analysis confirmed that adding 0.3% AOCP improved the NH_3_-N, BCP, and propionate concentrations in rumen fluid, and decreased the number of protozoa, acetate concentration, and acetate/propionate values [14]. It is believed that rumen microbiota play a significant role in reducing the inflammatory response of the host by inhibiting the level of certain cytokines, macrophage, and neutrophil proliferation [15]. Several studies have demonstrated that the addition of *Artemisia annua* extract (0.75%) improves rumen fermentation in dairy cows, and its promoting effects are intimately linked to changes in rumen microbial composition.

Nevertheless, it has not been published on the effect of AOCP on growth performance, nutrient digestibility, antioxidant and immunity capacity, rumen fermentation, and microbiota in Albas White cashmere goats. Based on the functions of antioxidants and the regulation of the immune and antioxidant activity of AOCP in other species of animals, as well as the modulation of ruminal fermentation and growth performance by *Artemisia annua* extract. We evaluated the effects of AOCP on the antioxidants, immunity, rumen fermentation, nutrient digestibility, bacterial communities, and production performance of Albas White cashmere goats to elucidate how AOCP affects the health of cashmere goats and to discuss the major microflora influencing the host’s health status in conjunction with macrobiotics, providing a basis for the utilization of *Artemisia ordosica* resources in the field of feeds as well as to propose new methods for developing alternative antibiotics in the breeding of cashmere goats.

## 2. Materials and Methods

The experiment was conducted at the animal breeding base of the College of Animal Science, Inner Mongolia Agricultural University, China (40.8° N and 111.7° E). The climate of the region is characterized by a temperate continental monsoon climate. Animal Ethics and Welfare Committee of Inner Mongolia Agricultural University approved the animal use following the Laboratory Animal Sciences and Technical Committee of the Standardization Administration of China, and performed under the national standard Guidelines for Ethical Review of Animal Welfare [16].

### 2.1. Preparation of Artemisia Ordosica Crude Polysaccharide

Fresh A. ordosica plants (aerial part) were collected from Erdos (40.41° N and 110.03 ° E, Inner Mongolia, China) in August. We crushed them, and converted them into a dry powder that provided the nutritional values shown in Table 1. A crude polysaccharide extraction from *Artemisia ordosica* was performed in our laboratory according to the method of Xing et al. [17]; the yield was 5.56%, and the sugar content was 52.65%. We used ion exchange chromatography (Dionex ICS 5000, Pudi Biotechnology Co., Ltd., Shanghai, China) equipped with a CarboPac PA20 analytical column (250 × 4 mm; Dionex) to identify the monosaccharide composition of AOCP. The results are shown in Table 2.

### 2.2. Experimental Animals, Diets, and Design

Twelve healthy, castrated male, 2-year-old cashmere goats (38.03 ± 2.42 kg of BW ± SD, with the strain of Inner Mongolia Albas white cashmere goat) were selected in a single-factor completely randomized design. As part of a 2 × 2 crossover design, two groups of goats were assigned to two treatments: a basal diet without supplementation (CON) or a supplemented diet containing 3 g/kg DM (0.3% AOCP), resulting in six replicates for each treatment. The goats were fed with total mixed ration (TMR). All ingredients were mixed in TMR, and 3 g/kg DM AOCP was mixed with 200 g of TMR, then top-dressed on the rest of TMR in the morning. All the goats were housed individually and fed ad libitum twice per day (09:00 h and 15:00 h) with free access to fresh water. The dietary concentrate/roughage ratio was 3:7, and the ingredients and nutrients levels are shown in Table 3. The diet provided in this study was carefully monitored to ensure that aflatoxin levels were well below the established safety limits for animal feed. Throughout the experiment, a humidity level of 60% to 70% and a temperature range of 25 °C to 27 °C were maintained. The experiment lasted for 28 days, consisting of 7 days of adaptation and 21 days of data and sample collection. The initial body weights were recorded on the first day of the experiment before the morning feeding to avoid weighing errors due to the variation in gut filling. Then, the cashmere goats were weighed weekly (7 days, 14 days, and 21 days) using a traditional balance (Foshan Shunde District Heng edge electronic technology Co., Ltd., Foshan, China), and the average daily weight gain was determined. In the course of the experiment, the cashmere goats’ average daily feed intake was calculated from the weight of the offered and remaining feed. Feed conversion ratios (FCRs) were calculated as total feed intake/body weight gain.

### 2.3. Sample Collection

Jugular blood samples were taken on day 28 of the experiment. Serum was collected from one goat per pen and centrifuged at 2500× *g* for 10 min at 4 °C using a Becton-Dickinson Vacutainer System (Franklin Lakes, NJ, USA). In addition, aliquots of serum were frozen immediately at −20 °C for subsequent analysis of their antioxidant capacity and cytokine indicators. Fecal output was recorded daily from days 22 through 28 after goats were fitted with a fecal collection bag. Feces samples were taken at 1/5 of the wet weight and stored at −20 °C. During the measurement period, samples were collected from eight signal days and then thawed and mixed. Approximately 200 g of feces were removed from each pen, mixed, dried for 72 h at 65 °C, and ground to pass a 1 mm screen. A sample of rumen fluid was obtained on day 28. Rumen fluid (100 mL) was collected from each goat using a tube-type nasogastric sampler (MDW15, Guidi Scientific Instrument Co., Ltd., Shanghai, China). Rumen fluid from the first two tubes was discarded in order to prevent contamination with saliva. A portable pH meter was used to determine the pH of each sample immediately following the collection of rumen fluid samples (A Yuan Instrument Co., Ltd., Suzhou, China). The rumen fluid was filtered through four layers of cheesecloth. Three aliquots of 1 mL samples were mixed with 4 mL of 3.5% formalin containing 8.0 g/L sodium chloride and 0.6 g/L methyl green and stored at 4 °C for the microscopic counting of protozoa; two aliquots of 0.5 mL filtrate were mixed with 4.5 mL of 0.2 mol/L hydrochloric acid to fix nitrogen for the determination of ammonia-N (NH_3_-N); two aliquots of 4 mL filtrate were mixed with 1 mL of 25% metaphosphoric acid solution for the determination of volatile fatty acids (VFAs); and the residual filtered liquid was collected for microbial protein (MCP) analysis. Samples for VFA, NH_3_-N, and MCP were stored at −20 °C until analysis. Three aliquots of 2 mL rumen fluid only collected at 1 h before morning feeding on 28 d were placed into cryogenic vials (Corning, New York, NY, USA), shock frozen in liquid nitrogen, and stored at −80 °C for the purpose of analyzing the composition of ruminal bacteria.

### 2.4. Chemical Analysis

Using the Association of Official Analytical Chemists (AOAC) methods, samples were analyzed in duplicate for dry matter (DM, NO.967.03), crude protein (CP, NO.954.01), and ether extract (EE, NO.920.39) [18], an ANKOM A2000i fiber analyzer (ANKOM Technology, New York, NY, USA) was used to determine neutral detergent fiber (NDF) and acid detergent fiber (ADF), which were both determined following the method described by Soest et al. [19]. A method for analyzing the concentrations of Ca and P was carried out via an inductively coupled plasma atomic emission spectrometry (ICP-AES) instrument (iCAP6300, Thermo Fisher, New York, NY, USA). The apparent total tract digestibility (ATTD) of a particular nutrient was calculated with acid-insoluble ash (AIA) according to Reilly et al. [20] based on the equation ATTD (%) = 100 − [(Crdiet × nutrient feces)/(Crfeces × nutrient diet)] × 100. The serum contents of total antioxidant capacity (T-AOC, U/mL), glutathione peroxidase (GSH, U/mL), superoxide dismutase (SOD, U/mL), catalase (CAT, U/mL), thioredoxin reductase (TrxR, U/mL), and malondialdehyde (MDA, nmol/mL) were measured in accordance using standard commercial kits (Nanjing Jiancheng Bioengineering Institute, Nanjing, China). The serum levels of interleukin (IL)-1β, IL-6, tumor necrosis factor (TNF)-α, nitric oxide (NO), inducible nitric oxide synthase (iNOS), and reactive oxygen species (ROS) were determined using ELISA test kits (Baoman Biological Technology Co., Ltd., Shanghai, China), following the manufacturer’s instructions.

### 2.5. Rumen Fermentation Index Measurement

It was determined that ammonia nitrogen (NH_3_-N), bacterial protein (BCP), and VFA concentrations were present based on colorimetric analysis by Miguel et al. [21], Coomassie brilliant blue analysis by Chanjula et al. [22], as well as gas chromatography analysis by Seo et al. [23]. DBA was used as the internal standard, and the treated solution (1 mL of rumen fluid from each goat was mixed with 200 μL of DBA for 10 min, then centrifuged at 3500× *g* and pipetted 1 mL of the supernatant) was injected into the sample bottle, and the procedure was started (column temperature of 120 °C, maintained for 1 min; ramped up to 180 °C at 30 °C/min; the final stage was maintained for 10 min at 180 °C) for gas chromatographic determination (Agilent 7890A, Palo Alto, CA, USA). Gas chromatography conditions were as follows: chromatographic column (DB-FFAP-122-3263, 60 m, 0.25 mm, 0.5 μm); inlet temperature: vapor chamber temperature was set at 220 °C; carrier gas was nitrogen; pressure was 193.1 kpa; total flow rate was 53.7 mL/min; injection volume was 1 mL; split ratio was 40:1. Regarding the detector temperature, the temperature of the detection chamber was monitored and determined to be 250 °C through a hydrogen flame ionization detector. A staining microscopy method based upon the method proposed by Yesilbag et al. [24] was used to determine the number of protozoa, and gas production was determined according to Brewster et al. [25], employing the ANKOMRFS in vitro gas production system (RFS, Sichun Anhao Zotye Technology Co., Ltd., New York, NY, USA).

### 2.6. Analysis of Microbial Community in Ruminal Fluid

An analysis of ruminal microbial community diversity was performed utilizing 16S rRNA gene sequencing. From the ruminal fluid samples, microbial community genomic DNA was extracted using a DNA kit (Omega Bio-tek, Norcross, GA, USA). Analyses were conducted using 1% agarose gels to determine the concentration and purity of the DNA extracts. An ABI GeneAmp^®^ 9700 PCR thermal cycler (ABI, Waltham, MA, USA) was used to amplify the bacterial 16S rRNA high-variant region V3–V4 using primer pairs 338F (5′-ACTCCTACGGGAGGCAGCAG-3′) and 806R (5′-GGACTACHVGGGTWTCTAAT-3′). As part of the process of PCR analysis, the PCR products were extracted from 2% agarose gels, purified using an AxyPrep DNA Gel Extraction Kit (Axygen Biosciences, Union City, CA, USA), and quantified using a QuantusTM fluorometer (Promega, Madison, WI, USA). Sequencing was conducted on the Illumina MiSeq PE300 platform/NovaSeq PE250 platform (Illumina, San Diego, GA, USA) with the purified amplification products. A clustering algorithm was developed by using UPARSE software with 97% similarity between operating units (OTUs) (version 7.0.1090, available at http://drive5.com/uparse/, (accessed on 14 May 2022)). Sequences of the 16S rRNA gene were classified using the ribosomal database project (RDP) classifier (version 2.13, https://sourceforge.net/projects/rdp-classifier/, (accessed on 14 May 2022)) using a 70% threshold for comparison. A calculation of alpha diversity at the OTU level was performed using Mothur software (version 1.30.2, available from http://www.mothur.org/wiki/Download_mothur, (accessed on 14 May 2022)). For the purpose of analyzing the differences between the two groups in terms of alpha diversity indices (Sobs, ACE, Chao, Shannon, and Simpson), the Wilcoxon rank sum test was employed. Qiime software was used to perform beta diversity analysis in order to determine the similarity between different samples of community structure based on the Bray–Curtis distance algorithm (version 1.91, http://qiime.org/install/index.html, (accessed on 14 May 2022)). For the purpose of observing the differences among the samples, principal coordinate analysis (PCoA) and non-metric multidimensional scaling analysis (NMDS) were performed to naturally decompose the community structure data. Differentiated bacteria were analyzed using linear effect size discriminant analysis (LEfSe). Based on the LEfSe coefficient, Wilcoxon rank tests were conducted to detect taxa with significant differences in abundance (http://huttenhower.sph.harvard.edu/galaxy/root?tool_id=lefse_upload, (accessed on 14 May 2022)). Spearman rank correlation coefficients were applied to analyze the Correlation Heatmap. As part of this study, a matrix of values was calculated to examine the relationship between environmental factors and the selected species. In a two-dimensional matrix or table, color changes represent information about the data, while the colors indicate the magnitude of the data values, which can be visualized using the defined colors. The data were processed and analyzed using the Majorbio Bio-Pharm Platform (Majorbio Bio-Pharm Technology Co., Ltd., Shanghai, China).

### 2.7. Statistical Analysis

The statistical significance of data was evaluated using SAS 9.2, using the Paired-Samples T Test procedure on normally distributed data, or otherwise using the Kruskal–Wallis test. Data are expressed as the least square mean and standard error of the mean. Kruskal–Wallis rank sums were utilized to analyze differences between phyla and families at the ruminal bacteria level. Differences in ruminal bacterial abundance of goats at the genus level were classified via LEfSe analysis if the LDA score of the ruminal microbiota exceeded 2. Significant differences were defined as *p*-value < 0.05, while 0.05 ≤ *p*-value < 0.10 were considered to have a trend towards significance.

## 3. Results

### 3.1. Growth Performance

The effects of AOCP on the growth performance in cashmere goats are presented in Table 4. It was concluded that there was no significant difference in initial body weight between goats fed CON and AOCP diets (*p* > 0.05). *Artemisia ordosica* crude polysaccharide supplemented with the ADG of goats was significantly greater than CON group (*p* < 0.05), whereas the F/G in the AOCP group was lower than the CON group (*p* < 0.05). No difference was observed for DMI among the two dietary treatments (*p* > 0.05).

### 3.2. Nutrient Digestibility

Table 5 illustrates the effect of AOCP on the apparent digestibility of nutrients. CP and ADF digestibility in goats were affected by adding AOCP diets (*p* < 0.05). No significant difference was found between the CON and AOCP treatment groups in terms of NDF, Ca, and P digestibility (*p* > 0.05). However, there was a tendency for the DMI levels in the AOCP group to be higher than in the CON group (*p* = 0.052).

### 3.3. Serum Oxidative Status

The results of the study are presented in Table 6. A comparison between the AOCP and CON groups revealed an increase in serum levels of CAT, GSH-Px, and T-SOD in the AOCP group (*p* < 0.05). No difference was observed for MDA, TrxR, and T-AOC among the two dietary treatments (*p* > 0.05).

### 3.4. Serum Inflammatory Cytokines

Table 7 summarizes the effects of AOCP on serum immune responses. Comparatively to the CON group, AOCP supplementation increased 1L-6 and NO serum concentrations in cashmere goats (*p* > 0.05). There were no significant differences in 1L-1β, ROS, TNF-α, and iNOS serum concentrations in goats offered the AOCP diets (*p* > 0.05).

### 3.5. Rumen Fermentation Characteristics

As shown in Table 8, the AOCP group significantly increased the levels of BCP in goats compared to the CON group (*p* < 0.05). It was observed that the protozoon population in the AOCP group was lower than that in the CON group (*p* < 0.05). In particular, there was no difference in the rumen pH and NH_3_-N concentrations between the two dietary treatments (*p* > 0.05). Table 9 illustrates the effects of AOCP on VFA concentrations. Compared to the CON group, the AOCP group markedly increased the concentrations of propionate, butyrate, iso-butyrate, valerate, and TVFA while decreasing the concentrations of acetate and A/P (*p* < 0.05). As far as iso-valerate is concerned, no difference was observed between the two dietary treatments (*p* > 0.05).

### 3.6. Rumen Bacterial Community Richness, Diversity, and Composition

The α-diversity analysis revealed that the Sobs (*p* = 0.001) and Ace (*p* = 0.002) indexes in the AOCP group were increased, the Simpson indexes were decreased (*p* < 0.001), and the Shannon indexes showed a tendency to rise (*p* = 0.077), which illustrated that the addition of AOCP increased the ruminal microbial community’s richness and diversity (Figure 1C). Further, the results showed that the coverage rate reached 99%, and that the change of rarefaction curves was generally slow and close to saturation, indicating that the microbial species and structural diversity of the rumen could be accurately reflected (Figure 1A). The clustering of nonrepetitive sequences on the basis of 97% similarity yielded 2243 OTUs. However, OTU Venn analysis indicated that there were 369 and 327 unique OTUs in the CON and AOCP diets, respectively (Figure 1B). Compared to the CON group, principal coordinate analysis (PCoA) using Bray–Curtis’s distance matrices showed that significantly higher variations were detected in rumen fluid samples from the AOCP group, and that PC1 and PC2 accounted for 30.78% and 11.65% of the total variation, respectively (Figure 2A). As shown by the NMDS plot analysis (stress = 0.076), there were significant differences between the CON and AOCP groups in terms of rumen microbial species and abundance (Figure 2B).

### 3.7. Significantly Different Ruminal Bacteria between the CON and AOCP Groups

As a phylum, Firmicutes and Bacteroidetes accounted for 90% of the microbial diversity (Table 10, Figure 3A,B). After supplementation with the AOCP diet, the relative abundance of Firmicutes decreased by 22.04% in comparison with the CON group. Conversely, the abundance of Bacteroides was reduced by 30.54% with the AOCP diet. It was observed that the majority of bacteria in the Firmicutes phyla consisted of *Prevotellaceae*, *Rikenellaceae*, *Bacteroidales_RF16_Group*, and *F082*, whereas *Ruminococcaceae*, *Oscillospiraceae*, *Lachnospiraceae*, *Selenomonadaceae*, *Christensenellaceae*, *UCG-010*, *norank_o_Clostridia_vadinBB60_group*, *Eubacterium_coprostanoligene_group*, *UCG-011*, and *norank_o_Clostridia_UCG-14* were members of the Bacteroidetes phyla (Figure 3B and Figure 4B). In turn, the relative abundance of *F082* was lower for the AOCP group than for the CON group (*p* < 0.05).

An LEfSe analysis was performed in order to identify different bacterial compositional differences between the two treatments of ruminal fluid in goats (Figure 5A,B). A total of 21 OTUs were obtained from 12 rumen fluid samples from goats. In the different OTUs, nine of them were assigned to the CON group, while twelve belonged to the AOCP group. Within the CON group, *Aeromonas*, *Ureibacillus*, *Megamonas*, *Streptococcus*, *Peptococcus*, *Lactococcus*, *norank_f_norank_o_Chloroplast*, *Escherichia Shigella*, and *unclassified_c_Alphaproteobacteria* were most prevalent. A great abundance of *norank_f_F082*, *Saccharofermentans*, *norank_f_norank_o_Bacteroidales*, *Anaerovibrio*, *Lachnospiraceae_FE2018_group*, *Ruminococcus gauvreauii group*, *Shuttleworthia*, *Eubacterium brachy group*, *norank_f_norank_o_Rickettsiales*, *Lachnospiraceae_UCG-009*, *norank_f__norank_o_Absconditabacteriales_SR1*, and *unclassified _p _Firmicutes* in the AOCP group was present based on the ranking of LDA values from high to low (Figure 5B).

An investigation was conducted into the inter-relationships between ruminal fermentation (Acetate et al.), growth performance (feed-to-gain ratio), serum oxidation status (GSH-Px), inflammatory status (1L-6), and the rumen microbiota. As a result of Spearman correlation coefficient analysis, several positive and negative correlations were found between rumen bacteria and ruminal fermentation, growth performance, serum oxidation, and inflammation status (Figure 5C). In the AOCP group, the relative abundance of *Lachnospiraceae_UCG-009* genera was negatively correlated with the concentration of acetate (r = −0.64, *p* < 0.05) and the F/G ratio (r = −0.59, *p* < 0.05), whereas the activity of GSH-Px (r = 0.61, *p* < 0.05) was positively correlated with it. The proportion of acetate (r = −0.67, *p* < 0.05) was negatively correlated of the genera *Ruminococcus_gauvreauii_group* and the F/G ratio (r = −0.62, *p* < 0.05). The relative abundance of *Shuttleworthia* genera was negatively correlated with acetate (r = −0.59, *p* < 0.05) and the F/G ratio (r = −0.66, *p* < 0.05). The ratio of F/G (r = −0.62, *p* < 0.05) and A/P (r = −0.60, *p* < 0.05) was negatively correlated of the genus *norank_f__F082* and had an extremely negative correlation with the concentration of acetate (r = −0.73, *p* < 0.01). However, the proportion of Butyrate (r = 0.63, *p* < 0.05) was positively correlated with the genera *norank_f__F082* and the activity of GSH-Px (r = 0.67, *p* < 0.05). The concentration of acetate (r = −0.69, *p* < 0.05) was negatively correlated with the genus *unclassified_p__Firmicutes* and the value of A/P (r = −0.63, *p* < 0.05), and extremely positively correlated with the activities of GSH-Px (r = 0.73, *p* < 0.01) and 1L-6 (r = 0.71, *p* < 0.01). The relative abundance of *Saccharofermentans* was negatively correlated with the ratios of the F/G (r = −0.69, *p* < 0.05) and A/P (r = −0.62, *p* < 0.05), and had an extremely negative correlation with the concentration of *acetate* (r = −0.72, *p* < 0.01). The ratio of F/G (r = −0,68, *p* < 0.05) and A/P (r = −0.60, *p* < 0.05) was negatively correlated in the genera *Anaerovibrio* and exhibited a highly negative correlation with the concentration of acetate (r = −0.73, *p* < 0.01), and was positively correlated with the activity of GSH-Px (r = 0.59, *p* < 0.01). The relative abundance of *Lachnospiraceae_FE2018_group* was negatively correlated with the concentration of acetate (r = −0.70, *p* < 0.05) and the values of F/G (r = −0.69, *p* < 0.05) and A/P (r = −0.59, *p* < 0.05). The ratio of F/G (r = −0.67, *p* < 0.05) was negatively correlated with the genus *norank_f__norank_o__Rickettsiales* and the proportion of acetate (r = −0.67, *p* < 0.05), and positively correlated with the concentration of TVFA (r = 0.47, *p* < 0.05). The concentration of acetate (r = −0.59, *p* < 0.05) was negatively correlated of the genera *norank_f__norank_o__Bacteroidales* and the ratio of A/P (r = −0.58, *p* < 0.05), whereas the proportion of propionate (r =0.61, *p* < 0.05) and butyrate (r = 0.66, *p* < 0.05) was positively correlated with it. The relative abundance of *Eubacterium_brachy_group* was negatively correlated with the concentration of acetate (r = −0.58, *p* < 0.05). On the other hand, in the CON diet, the concentration of butyrate was negatively correlated with the genera *Ureibacillus* (r = −0.61, *p* < 0.05) and *Megamonas* (r = −0.58, *p* < 0.05). The relative abundance of *Escherichia-Shigella* was extremely negatively correlated with the concentration of NH_3_-N (r = −0.76, *p* < 0.01) and negatively correlated with the activity of GSH-Px (r = −0.65, *p* < 0.05). The activity of GSH-Px (r = −0.58, *p* < 0.05) was extremely negatively correlated with that of the genera *Lactococcus*. The relative abundance of the genera *Aeromonas* was positively correlated with the concentration of acetate (r = 0.64, *p* < 0.05). The concentration of NH_3_-N (r = −0.70, *p* < 0.05) and TVFA (r = −0.70, *p* < 0.05) was negatively correlated with the genus *Streptococcus* and the activity of GSH-Px (r = −0.59, *p* < 0.05). The relative abundance of *unclassified_c__Alphaproteobacteria* was positively correlated with the ratios of F/G (r = −0.58, *p* < 0.05) and A/P (r = −0.58, *p* < 0.05), and extremely positively correlated with the concentration of acetate (r = 0.74, *p* < 0.05). In addition, the concentration of butyrate (r = −0.70, *p* < 0.05), propionate (r = −0.63, *p* < 0.05), and 1L−6 (r = −0.58, *p* < 0.05) was negatively correlated with the genus *unclassified_c__Alphaproteobacteria* and extremely negatively correlated with the concentration of TVFA (r = −0.74, *p* < 0.01). The ratios of F/G (r = 0.58, *p* < 0.05) and A/P (r = 0.68, *p* < 0.05) were positively correlated in the genera *Peptococcus* and exhibited highly positive correlation with the concentration of acetate (r = 0.75, *p* < 0.01). Furthermore, the proportion of propionate (r = −0.64, *p* < 0.05) was negatively correlated with the genus *Peptococcus* and extremely negatively correlated with the concentration of 1L-6 (r = −0.73, *p* < 0.01).

## 4. Discussion

As demand increased for mutton as well as the degradation of grassland ecosystems and the limited pasture capacity, including housing, feeding, and fattening cashmere goats, have emerged as new sources of economic growth. However, with the development of intensive livestock and poultry breeding, metabolic disorders and increased free radicals in animals have become more widespread. Due to these disorders, the antioxidant defense system becomes damaged, resulting in a variety of diseases [26]. Antioxidant enzymes are effective due to their ability to scavenge free radicals generated during metabolism. It is well established that all aerobically metabolized cells contain GSH-Px and SOD, which protect them from free radical damage and provide a mechanism for repairing oxidized cell membranes [27]. The accumulation of free radicals can lead to an immune imbalance if they are not promptly eliminated. It is possible that this will result in the release of inflammatory mediators and inflammatory responses in the body. However, immune disorders can aggravate oxidative stress, resulting in a vicious cycle. Consequently, this results in decreased animal health and economic productivity [28]. Inflammatory cytokines such as 1L-6 and NO are major factors responsible for altered metabolic cascades following the innate immune response and play an essential role in inflammatory responses [29]. Under mild oxidative stress, the concentration of pro-inflammatory factors is increased, allowing the immune system to be activated more rapidly and to resist disease and stress more efficiently [30]. According to Xing et al. [14], supplementing Arbor Acres male broilers with *Artemisia ordosica* polysaccharide (750 mg/kg) alleviated LPS-induced decreases in hepatic antioxidant enzyme activity (T-AOC, SOD, CAT, and GSH-Px) and increases in inflammatory cytokines (IL-6 and IL-2) by modulating the Nrf2/Keap1 and TLR4/NF-κB signaling pathways. It is consistent with the results of our experiment, which demonstrated that adding crude polysaccharides of *Artemisia ordosica* to the diet of cashmere goats increased serum T-SOD, CAT, GSH-Px, 1L-6, and NO concentrations while decreasing serum MDA and ROS contents, indicating that the addition of AOCP improved the cashmere goat’s resistance to external stressors by enhancing the activity of antioxidant enzymes and the release of inflammatory substances. Additionally, Yang et al. [31] demonstrated that *Artemisia argyi* flavonoids significantly reduced the decrease in average daily gain (ADG) at 21 and 35 days following the application of lipopolysaccharide (LPS, 500 μg/kg) by modulating the Keap1/Nrf2-ARE and NF-κB pathways, enhancing antioxidant enzyme function, and decreasing the release of inflammatory factors. Researchers have discovered a strong connection between rumen fermentation metabolites and the host’s immunological and antioxidant systems [32]. Several extracellular signal-regulated kinases 1/2 and p38 mitogen-activated protein kinases are thought to be activated by GPR41 and GPR43 in epithelial cells. However, acetate, propionate, and butyrate have been shown to have a close relationship with the SCFA receptors GPR41, GPR43, and GPR109A [33]. These substances alleviate the body’s inflammatory response by inhibiting neutrophils and macrophages while enhancing chemokines and cytokines [34]. As reported by Yu et al. [35,36], *Artemisia annua* modulates the composition of VFAs and improves antioxidant activity in goats. In our study, AOCP decreased the protozoa population in rumen fluid. It also increased the concentrations of NH_3_-N and BCP, as well as propionate, butyrate, isobutyrate, valerate, and TVFA in the fluid, while also reducing the acetate content and the acetate-to-propionate ratio in the rumen fluid. Similarly, Faryabi’s research on lambs found a significant decrease in rumen fluid acetate concentrations, an increase in propionate concentrations, and an improvement in blood antioxidant levels after using 25% *Artemisia absinthium* leaves as opposed to alfalfa hay [37]. According to the above results, AOCP enhances the body’s immune and antioxidant functions through its effect on the levels of VFAs. A probiotic effect of AOCP may also be responsible for this discovery since it affects the rumen microbial composition of cashmere goats. As a consequence, its metabolites (acetate, propionate, and butyrate) can activate the relevant signaling pathways, increasing the organism’s antioxidant and immune functions.

Several studies have confirmed that samples are adequately sampled if the sample coverage is greater than 97% [38]. In this experiment, a sample coverage of over 99 percent was attained, showing that the sequencing data are representative of the species and structural diversity of rumen microbes in cashmere goats. The a-diversity can be used as an index of the functional resilience of the gut microbial ecosystem, including species richness (Sobs, Chao, and Ace) and species diversity (Shannon, Simpson, and coverage), with all indices except the Simpson index correlating positively with rumen microbial diversity [39]. Following the addition of crude polysaccharides from *Artemisia ordosica*, the Simpson index decreased significantly, while the Ace index and Sobs index increased significantly. Consequently, supplementation with the crude polysaccharides of *Artemisia ordosica* increased the diversity and richness of the rumen microbial populations. As displayed by the PCoA and NMDS plots, it was evident that the consumption of AOCP resulted in the appearance of a different type of rumen bacteria.

Numerous studies have demonstrated that Bacteroidetes and Firmicutes are the predominant rumen microorganism strains in ruminants [40]. It is primarily Bacteroidetes that breakdown non-fiber carbohydrates in the diet, while Firmicutes are primarily responsible for breaking down fibrous carbohydrates in the rumen. As a consequence of the fermentation process in the rumen, cellulose and non-fibers produce acetate and propionate [41]. Following addition of the crude polysaccharides from *Artemisia ordosica*, it was found that the relative abundances of Firmicutes decreased while those of *Bacteroidetes* increased. Concomitantly, its proportion was higher than that of *Firmicutes*, suggesting that crude polysaccharides from *Artemisia ordosica* inhibited the growth and reproduction of *Firmicutes* microorganisms. *Synergistota* are in charge of the protein and carbohydrate degradation, which has an effect on the host’s level of SCFA [42]. In contrast, SCFAs are powerful anti-inflammatory molecules that are commonly found in intestinal bacteria. They have an essential role in immunomodulation by reducing immune cell adhesion and chemotaxis, increasing the release of anti-inflammatory cytokines, and inducing apoptosis, all of which have an anti-inflammatory effect on the host [43]. According to a recent study, the Proteobacteria phylum includes pathogenic bacteria such as *Escherichia coli* and *Salmonella*, which are capable of causing microbial dysbiosis and increasing disease risks in animals [44,45]. Nevertheless, supplementation with crude polysaccharides from Artemia ordosica decreased ruminal *Proteobacteria* phyla abundance in goats, which was associated with a decrease in intestinal disease. Based on the family level, a decrease in the relative abundance of *Selenomonadaceae* was observed by the AOCP group. An increase in the relative abundance of *Lachnospiraceae*, *F082*, and *Rikenellaceae* was noted by the CON group. This is due to *F082* having a larger number of members than the CON family in rumen fluid. By identifying unique high-dimensional biomarkers for microbial community analysis, LEfSe analysis confirmed these findings [46]. It has been noted that *norank_f__F082* belongs to *Bacteroidetes*, which is mainly involved in the degradation of non-cellulose, with propionate and butyrate as its primary metabolites [47]. The results of the present study indicate that *norank_f__F082* positively correlates with butyric acid concentration and GSH-Px activity in goats and negatively correlates with F/G. This indicates that this genus influences GSH-Px activity and improves feed conversion efficiency by stimulating butyric acid secretion in goats. As members of the family *Rickettsiaceae*, *Norank_f_norank_o_Rickettsiales* are not necessarily harmful to cashmere goats’ health. In contrast, they may transmit zoonotic diseases if excreted with feces [48]. According to research data, propionate has the ability to improve the conversion and storage of glucose, while butyrate provides energy to all tissues in the body [49]. The *norank_f__norank_o__Bacteroidales* and *Saccharofermentans* genera consume cellulose and hemicellulose, producing large amounts of propionate [50,51]. The *norank_f__norank_o__Bacteroidales* genus also showed a substantial positive correlation with propionate concentration, as well as a significant negative correlation with the acetate proportion and A/P ratio. Thus, this bacterium is capable of causing a significant reduction in A/P values by being metabolically active in the formation of acetate and propionate. Additionally, the *norank_f__norank__o_Bacteroidales* genus showed a negative connection with F/G values, which suggests that this bacterium enhances feed conversion efficiency in goats through changing rumen microbial composition and its metabolites by activating immune-related pathways. Microorganisms of the genera *Lachnospiraceae_FE2018_group* and *Eubacterium_brachy_group* play an instrumental role in energy homeostasis, immune regulation, and intestinal inflammation. By degrading non-fibrous carbohydrates (starch and sugar), it produces short-chain fatty acids, such as butyrate, which in turn can convert primary bile acids into secondary bile acids, preventing *Clostridium* perfringens infection [52,53,54]. Furthermore, there was a negative correlation found between the *Lachnospiraceae_FE2018_group* genus and F/G, suggesting that it promotes productive performance through the regulation of immunity. *Rumenococcus* degrades cellulose, hemicellulose, and lignin in roughage in the rumen, producing large quantities of cellulase and xylanase and generating acetate enzymatically [55]. A significant negative correlation was found between *Rumenococcus* concentration and acetate concentration in this experiment, indicating that *Rumenococcus* promoted the metabolism of acetate and had a substantial impact on the A/P value, which also indicated why acetate concentrations were significantly reduced in this study. Overall, the relative abundance of the above bacteria increased significantly after adding *Artemisia ordosica* polysaccharide to the diet. This may be partly responsible for the substantial increase in propionate and butyrate concentrations in the rumen fluid of goats. The *unclassified_p__Firmicutes* belong to the Firmicutes phylum and are associated with the degradation of fiber, immune regulation, and the deposition of fat in the rumen [56]. In the present study, the *unclassified_p__Firmicutes* genus was negatively correlated with acetate concentration and positively correlated with GSH-Px and IL-6. Compared to the CON group, the AOCP group showed an increase in ADG of 53.62% and a decrease in the F/G ratio of 32.38%. Nevertheless, there was no significant difference in the DMI between the two groups, suggesting that the increase in body weight was not due to the amount of food consumed. It is speculated that the reason may be that the relative abundance of the *unclassified_p__Firmicutes* genus increased, which has a positive effect on goat performance by regulating antioxidant and immune functions. Similar findings have been reported by Christine et al. [57]. In addition, the increase in relative abundance of these genera of bacteria led to the generation of short-chain fatty acids (propionate, butyrate, and acetate), which facilitated the degradation of cellulose in the feed, thereby improving the digestibility and availability of nutrients. It is in agreement with the findings of the study that the addition of crude polysaccharides from *Artemisia ordosica* significantly improved the digestibility of calcium and ADF in cashmere goats in this experiment, which would be beneficial for the fattening of cashmere goats. Researchers have hypothesized that the *Megamonas* genus is associated with the physiological process of immune disease and that their metabolism can result in the production of butyrate [58]. Both the *Aeromonas* and *Escherichia-Shigella* genera are ubiquitous pathogenic bacteria, among which *Aeromonas* can cause disease in aquatic animals, and *Escherichia-Shigella* has caused gastrointestinal infections or inflammation [59,60]. Both of the above genera are members of the CON group, which suggests that crude polysaccharides produced by *Artemisia ordosica* may have beneficial effects on ruminal function and structure. Nevertheless, the reason why the crude polysaccharides of *Artemisia ordosica* can exert this effect may be closely related to its active substance. The crude polysaccharide belongs to polyphenols, and the hydroxyphenyl structure in polyphenols causes it to have some antibacterial effects [61]. In this regard, it is possible that the antibacterial effect of polysaccharides may contribute to the reduction in the relative abundance of potentially pathogenic bacteria in rumen fluid. There was a negative correlation between the TVFA concentration in the CON group and *Streptococcus* and *unclassified_c__Alphaproteobacteria* genera, indicating that these two bacteria inhibited the degradation of soluble carbohydrates [62,63], which also explained why TVFA concentrations in the CON and AOCP groups were lower. Polysaccharides have prebiotic properties, and their beneficial effects are mainly seen in the composition of rumen microflora and their metabolite SCFAs. Therefore, the supplementation of AOCP may have affected the rumen microbial composition, and its metabolite SCFAs improved the growth performance of cashmere goats through the activation of antioxidant- and immune-related pathways. However, the underlying mechanism behind this process needs further research.

Numerous studies have demonstrated that *Artemisia* plant extracts have a positive effect on animal growth performance. At present, studies on crude polysaccharides from *Artemisia ordosica* are mainly focused on mice, pigs, and poultry, and studies on goats have not been reported. Weaned piglets’ apparent digestibility of nutrients and feed conversion efficiency were markedly improved by a water extract of *Artemisia ordosica* (750 mg/kg) in a dose-dependent manner, as well as considerably reducing their DMI from 14 to 28 days [10]. According to our study, AOCP improved the digestibility of DM, CP, and ADF, and increased ADG and feed conversion ratios in cashmere goats. Growth performance is influenced by feed palatability, which is the basis for increased feed intake. Feed ingredients must be palatable to ensure the host consumes sufficient nutrients to maintain normal growth and production [64]. Herbal additives have been found to improve feed odor, resulting in a higher intake of feed [65]. According to our findings, there was no significant difference in DMI after adding AOCP to cashmere goat diets, recommending that AOCP addition does not affect palatability. Similar results have been reported by Kim et al. [66] in sheep. However, the improved growth performance may be attributed to changes in the rumen microflora. It was found that the *norank__f__F082*, *norank__f__norank__o___Bacteroidales* and *unclassified__p__Firmicutes* genera in the AOCP group were negatively correlated with F/G. Accordingly, the active ingredients within AOCP improved the digestion and absorption of nutrients in the gastrointestinal tract, which enhanced growth performance. These findings are similar to those of Kim et al. [67], who found that supplementation with *Lactobacillus*-fermented *Artemisia* princeps (5.0 g/kg) decreased the concentrations of intestinal *Salmonella* spp. and increased the body weight gain and feed efficiency of Hy-line Brown male chickens. However, further investigation is required to determine whether improved growth performance is related to improved palatability. In conclusion, the possible reasons for the growth-promoting effect of AOCP in this experiment are as follows: Firstly, it promoted the colonization of beneficial bacteria by positively influencing GSH-Px and IL-6 (*norank_f__F082* and *unclassified_p__Firmicutes*), as well as bacteria negatively associated with F/G (*norank_f__norank_o__Bacteroidales*, *unclassified_p__Firmicutes*, and *norank_f__F082*), and metabolites modulating immune competence, which in turn had a positive effect on goat health. Secondly, it is necessary to reduce the colonization of potential pathogenic bacteria (*Aeromonas* and *Escherichia-Shigella*). Lastly, cashmere goat growth performance can be improved through increased nutrient absorption and protein synthesis.

## 5. Conclusions

By adding 0.3% crude polysaccharide of *Artemisia ordosica* to the ration, the cashmere goats displayed improved growth performance, nutrient digestibility, antioxidant activity, immune response, and rumen fermentation. This phenomenon may be explained by the increased diversity and altered structure of the rumen microflora. As a result of AOCP supplementation, the colonization of beneficial bacteria has improved, including *norank__f__F082*, *norank__f__norank__o___Bacteroidales*, *Lachnospiraceae_FE2018_group*, and *unclassified__p__Firmicutes*, while the colonization of *Aeromonas*, *Escherichia-Shigella*, and other pathogenic bacteria has decreased.

## Figures and Tables

**Figure 1 animals-13-03575-f001:**
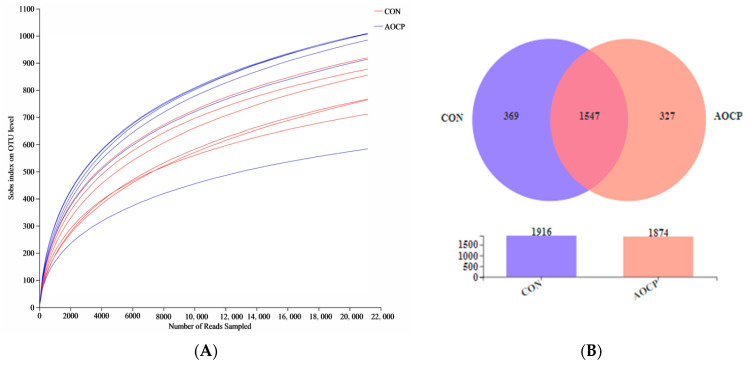
The ruminal fluid richness and diversity in CON and AOCP groups at (**A**) Rarefaction curve. (**B**) OTU Venn of two diets. (**C**) Comparison of a-diversity indices of two dietary treatments. ab Means in two groups that do not have a common marked letter differ significantly (*p* < 0.05).

**Figure 2 animals-13-03575-f002:**
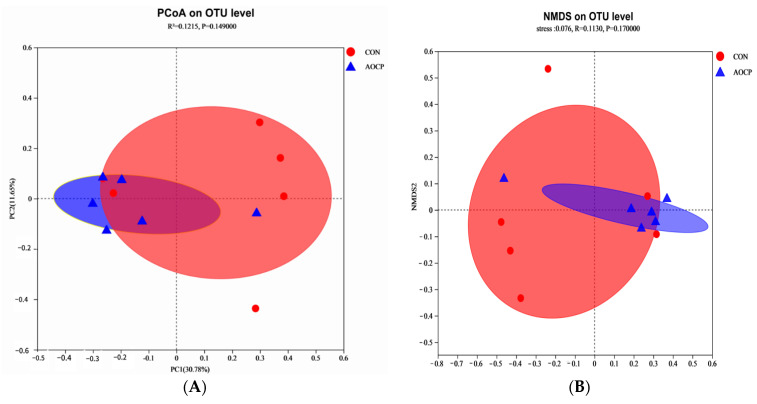
Beta diversity analysis of ruminal microbiota through (**A**) principal coordinate analysis (PCoA) and (**B**) non-metric multidimensional scaling analysis (NMDS).

**Figure 3 animals-13-03575-f003:**
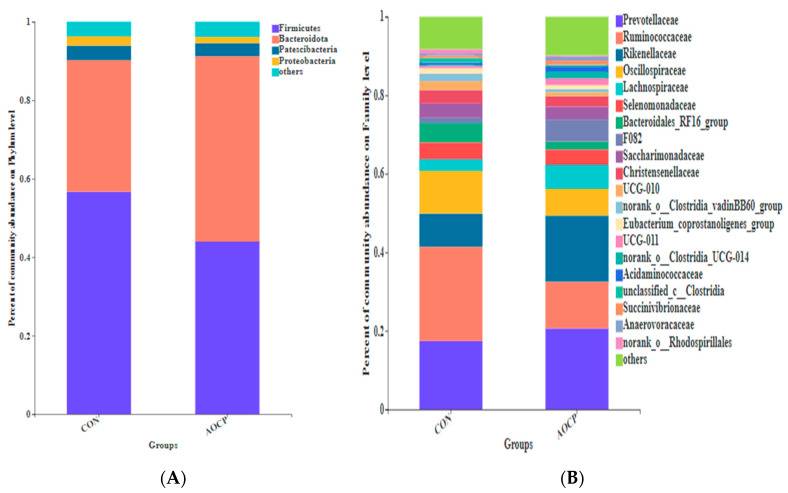
Relative abundance of ruminal microbiota at the (**A**) phylum and (**B**) family levels.

**Figure 4 animals-13-03575-f004:**
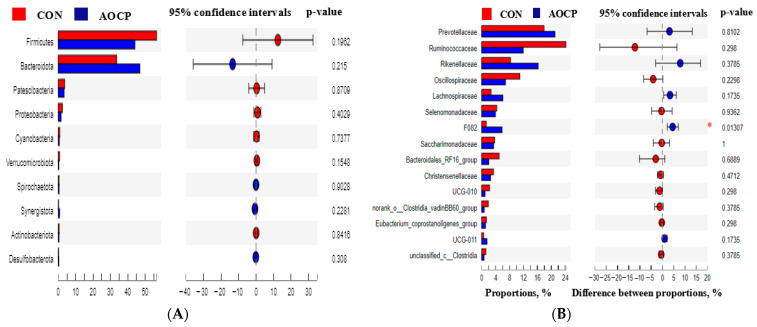
Differences in the rumen microbiota compositions at the (**A**) phylum level based on a contribution degree at top 10 and (**B**) family level based on a contribution degree at top 15. * *p* < 0.05.

**Figure 5 animals-13-03575-f005:**
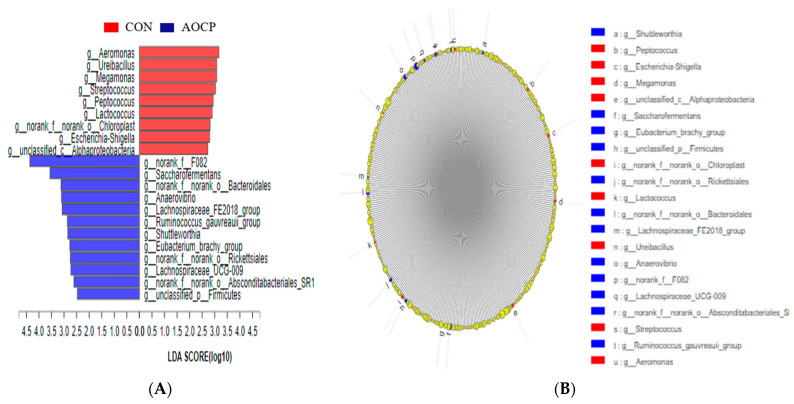
LefSE analysis of ruminal microbiota among two dietary treatments. (**A**) Linear discriminant analysis (LDA) score of the rumen microbiota, and a score ≥ 2 means significant. (**B**) Cladogram of LEfSe shows taxonomic profiling from the genus level. The yellow node represents no difference, but other color nodes represent significant difference. (**C**) Spearman′s correlation analysis between F/G, ruminal fermentation, serum oxidative and inflammatory status, and ruminal microbiota by LefSE analysis of ruminal microbiota from genus level. Only genera with *p* ≤ 0.01 were concerned (* 0.01 < *p* ≤ 0.05, ** 0.001 < *p* ≤ 0.01).

**Table 1 animals-13-03575-t001:** Nutrient composition and content of *Artemisia ordosica* (%, DM basis).

Nutrient Composition	Content, %
Crude protein	9.03
Ether extract	7.63
NDF	48.56
ADF	32.01
Calcium	1.14
Phosphorus	0.18

**Table 2 animals-13-03575-t002:** Monosaccharide composition and molar ratio of AOCP.

Items	AOCP
Arabinose	6.87
Galactose	10.67
Glucose	54.13
Xylose	2.49
Mannose	18.37
Galacturonic acid	4.83
Glucuronic acid	2.64

**Table 3 animals-13-03575-t003:** Composition and nutrient levels of the basal diet (air-dry basis).

Item	Content
Ingredient, g/kg air dry basis	
Millet straw	58.80
Alfalfa hay	2.96
Tall oat grass	8.08
Corn	14.72
Soybean meal	5.30
Distillers dried grains with solubles	3.30
Linseed cake	5.30
Limestone	0.12
CaHPO_4_	0.12
Premix	0.50
NaCl	0.30
NaHCO_3_	0.50
Total	100.00
Nutrient composition	
Digestible energy (MJ/Kg)	10.95
Crude protein	10.04
Ether extract	2.24
Neutral detergent fiber	54.71
Acid detergent fiber	30.98
Calcium	0.63
Phosphorous	0.29

One kilogram of the premix contained: vitamin A, 1200,000 IU; vitamin D3, 500,000 IU; vitamin E, 5000 IU; vitamin K3, 360 mg; vitamin B1, 70 mg; vitamin B2, 1700 mg; vitamin B6, 180 mg; nicotinic acid, 4.4 g; D-pantothenic acid, 3.4g; VB12, 6mg; biotin, 28 mg; folic acid, 300 mg; Fe, 8g; Cu, 1.6 g; Zn, 10 g; Mn, 6g; I, 60 mg; Se, 60 mg; Co, 50 mg. Digestible energy was a calculated value, while the others were measured values.

**Table 4 animals-13-03575-t004:** Effects of 0.3% *Artemisia ordosica* crude polysaccharide (AOCP) on growth performance of cashmere goats.

Item	CON	AOCP	SEM	*p*-Value
Initial BW, kg	38.54	38.15	0.870	0.769
ADG, g/d	37.43 ^a^	57.50 ^b^	11.711	0.007
DMI, g/d	1201	1248	97.522	0.715
F/G	32.09 ^a^	21.70 ^b^	6.634	0.016

AOCP = *Artemisia ordosica* crude polysaccharide; SEM = standard error of the mean. ^ab^ Means in a row that do not have a common superscript letter differ significantly (*p* < 0.05). Date are least squares means of six observations (pen as the experimental unit; single column feeding) per treatment.

**Table 5 animals-13-03575-t005:** Effects of 0.3% *Artemisia ordosica* crude polysaccharide (AOCP) on nutrient apparent digestibility of cashmere goats (%).

Item	CON	AOCP	SEM	*p*-Value
DM	85.46	86.64	0.316	0.052
CP	79.13 ^a^	81.58 ^b^	0.555	0.022
EE	81.60	80.41	0.353	0.079
NDF	70.96	71.76	0.270	0.148
ADF	67.57 ^a^	69.73 ^b^	0.365	0.016
Ca	56.58	62.00	2.732	0.262
P	71.34	69.64	1.043	0.325

AOCP = *Artemisia ordosica* crude polysaccharide; SEM = standard error of the mean. ^ab^ Means in a row that do not have a common superscript letter differ significantly (*p* < 0.05). Date are least squares means of six observations (pen as the experimental unit; single column feeding) per treatment.

**Table 6 animals-13-03575-t006:** Effects of 0.3% *Artemisia ordosica* crude polysaccharide (AOCP) on serum antioxidant of cashmere goats.

Item	CON	AOCP	SEM	*p*-Value
MDA, nmol/mL	2.85	2.78	0.204	0.819
CAT, U/mL	5.18 ^a^	6.39 ^b^	0.172	0.006
GSH-Px, U/mL	36.93 ^a^	45.97 ^b^	2.323	0.036
TrxR, U/mL	22.45	21.11	1.053	0.375
T-AOC, U/mL	2.57	2.48	0.217	0.173
T-SOD, U/mL	20.16 ^a^	22.34 ^b^	0.537	0.039

AOCP = *Artemisia ordosica* crude polysaccharide; SEM = standard error of the mean. ^ab^ Means in a row that do not have a common superscript letter differ significantly (*p* < 0.05). Date are least squares means of six observations (pen as the experimental unit; single column feeding) per treatment.

**Table 7 animals-13-03575-t007:** Effects of 0.3% *Artemisia ordosica* crude polysaccharide (AOCP) on serum immune response of cashmere goats.

Item	CON	AOCP	SEM	*p*-Value
IL-1β, pg/mL	35.54	39.72	1.953	0.199
IL-6, pg/mL	44.41 ^a^	60.88 ^b^	1.439	0.003
ROS, pg/mL	37.03	36.90	0.695	0.922
TNF-α, pg/mL	40.69	41.17	0.595	0.687
NO, µmol/L	7.17 ^a^	9.40 ^b^	0.299	0.002
iNOS, U/mL	3.21	3.64	0.209	0.421

AOCP = *Artemisia ordosica* crude polysaccharide; SEM = standard error of the mean. ^ab^ Means in a row that do not have a common superscript letter differ significantly (*p* < 0.05). Date are least squares means of six observations (pen as the experimental unit; single column feeding) per treatment.

**Table 8 animals-13-03575-t008:** Effects of 0.3% *Artemisia ordosica* crude polysaccharide (AOCP) on ruminal fermentation parameters of cashmere goats.

Item	CON	AOCP	SEM	*p*-Value
pH	6.96	6.90	0.071	0.603
NH_3_-N, mg/100 mL	18.93	20.33	1.009	0.369
BCP, mg/100 mL	34.43 ^a^	53.42 ^b^	3.878	0.030
Protozoon, ×10^4^/mL	10.84 ^a^	5.64 ^b^	3.750	0.004

AOCP = *Artemisia ordosica* crude polysaccharide; SEM = standard error of the mean. ^ab^ Means in a row that do not have a common superscript letter differ significantly (*p* < 0.05). Date are least squares means of six observations (pen as the experimental unit; single column feeding) per treatment.

**Table 9 animals-13-03575-t009:** Effects of 0.3% *Artemisia ordosica* crude polysaccharide (AOCP) on VFA of cashmere goats (mmol/L).

Item	CON	AOCP	SEM	*p*-Value
Acetate	35.95 ^b^	28.32 ^a^	1.759	0.025
Propionate	7.87 ^a^	12.23 ^b^	0.333	0.004
Butyrate	7.84 ^a^	10.39 ^b^	0.487	0.015
Iso-butyrate	1.08 ^a^	1.24 ^b^	0.038	0.021
Valerate	0.69 ^a^	0.80 ^b^	0.021	0.022
Iso-valerate	1.36	1.51	0.243	0.341
TVFA	47.18 ^a^	64.76 ^b^	1.364	0.004
A/P	4.63 ^b^	2.32 ^a^	0.204	<0.001

AOCP = *Artemisia ordosica* crude polysaccharide; SEM = standard error of the mean. ^ab^ Means in a row that do not have a common superscript letter differ significantly (*p* < 0.05). Date are least squares means of six observations (pen as the experimental unit; single column feeding) per treatment.

**Table 10 animals-13-03575-t010:** Effects of *Artemisia ordosica* polysaccharide on relative abundance of rumen microflora (phylum level) (%).

Item	CON	AOCP	SEM	*p*-Value
Firmicutes	56.53	44.07	6.363	0.224
Bacteroidota	33.76	47.20	7.135	0.242
Patescibacteria	3.67	3.34	1.400	0.754
Proteobacteria	2.39	1.63	0.608	1.000
Cyanobacteria	0.90	0.78	0.486	0.936
Verrucomicrobiota	0.85	0.37	0.216	0.128
Spirochaetota	0.51	0.54	0.188	0.894
Synergistota	0.22	0.84	0.343	0.470
Actinobacteriota	0.56	0.50	0.212	0.631
other	0.61	0.91	0.190	0.376

AOCP = *Artemisia ordosica* crude polysaccharide; SEM = standard error of the mean. Date are least squares means of six observations (pen as the experimental unit; single column feeding) per treatment.

## Data Availability

The data presented in this study are available upon request from the corresponding author.

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
