# Peer review of "Effects of Artemisia ordosica Crude Polysaccharide on Antioxidant and Immunity Response, Nutrient Digestibility, Rumen Fermentation, and Microbiota in Cashmere Goats"

_animals, 2023, doi:10.3390/ani13223575_

Round 1
Reviewer 1 Report
Comments and Suggestions for Authors
Dear authors,
I have reviewed your manuscript. The proposal is interesting and the research is very complete, good introduction and discussion and correct methodology in general terms.
Below, I make some observations and recommendations that I hope will be useful:
- Scientific names in italics
- Carefully check spaces between words
- Use superscripts in the table footnotes and for statistical differences
- Use subscripts in chemical formulas (NH3-N, for example)
- Improve the quality of the figures
Line 107-116: I recommend making a table to describe the composition of Artemisa ordosica and the polysaccharides
Line 117-121: I recommend mentioning in more detail the way in which the A. ordosica polysaccharides were supplemented.
Table 1: Use the word vitamin instead of V
Line 137: Are you referring to folic acid when you mention olic acid?
Line 141-157: 2.2 and 2.3 are the same. Delete duplicate.
Line 251-259: The use of Excel for statistical analysis is not common. Justify its use for this research.
Table 2: Include final weight and use DMI instead of ADFI.
Line 650-660: I recommend rewriting the conclusions so that they do not look like results.
Kind regards
Comments on the Quality of English LanguageI noticed some grammatical errors that can be easily fixed.
Author Response
Response to Reviewer 1 Comments
Dear Reviewer,
Thank you very much for your precious comments and advice. Those comments are all valuable and very helpful for revising and improving our paper, as well as the important guiding significance to our researches. We have studied comments carefully and revised our manuscript based on them. The detailed responds to your review are attached below.
We would love to thank you for allowing us to resubmit a revised copy of the manuscript and we highly appreciate your time and consideration.
Kind regards,
Authors
Dear authors,
I have reviewed your manuscript. The proposal is interesting and the research is very complete, good introduction and discussion and correct methodology in general terms.
Below, I make some observations and recommendations that I hope will be useful:
au: Thank you for your recognition and positive comments on this study. We will do our best to address your concerns in order to make the manuscript more complete and informative.
- Scientific names in italics
au: Thank you for your suggestion. Based on your comments, we have revised and highlighted the entire article. Please refer to the manuscript for details.
- Carefully check spaces between words
au: Thanks for your careful question. Based on your comments, we have revised and highlighted the entire article. Please refer to the manuscript for details.
- Use superscripts in the table footnotes and for statistical differences
au: Thank you for your suggestion. Based on your comments, we have revised and highlighted the entire article. Please refer to the manuscript for details.
- Use subscripts in chemical formulas (NH3-N, for example)
au: Thank you for your suggestion. Based on your comments, we have revised and highlighted the entire article. Please refer to the manuscript for details.
- Improve the quality of the figures
au: Thanks for your comments. Based on your suggestion, all figures are re-downloaded from the system in a format (SVG) appropriate for publication in the article. Please refer to the manuscript for details.
- Line 107-116: I recommend making a table to describe the composition of Artemisa ordosica and the polysaccharides
au: Thank you for your suggestion. According to your suggestion, use the table to describe the polysaccharides and the composition of Artemisia ordosica. Please refer to lines 118-120 for details. Further, we have corrected the description of Chapter 2.1 and the numbering of the corresponding tables.
- Line 117-121: I recommend mentioning in more detail the way in which the A. ordosica polysaccharides were supplemented.
au: Thank you for your suggestion. The supplement has been completed according to your suggestion. Please refer to lines 126-128 for details.
- Table 1: Use the word vitamin instead of V
au: Thanks for your question. The supplement has been completed according to your suggestion. Please refer to chapter 2.2 for details. Revisions have been made based on your recommendations. Please refer to the manuscript lines 143-144 for more information.
- Line 137: Are you referring to folic acid when you mention olic acid?
au: Thanks for your question. It should be folic acid. Please accept our apologies for any inconvenience our oversight may have caused. Please refer to line 145 for more details.
- Line 141-157: 2.2 and 2.3 are the same. Delete duplicate.
au: Thank you for your suggestion. The sequence number correction has been completed, and chapter 2.3 has been deleted. Please accept our apologies for any inconvenience caused by our oversight. Please consult the manuscript for details.
- Line 251-259: The use of Excel for statistical analysis is not common. Justify its use for this research.
au: Thank you for your suggestion. Admittedly, we are using Excel to organize the data. The data was analyzed using SAS 9.2, and corrections have been made. Please accept our apologies for any inconvenience caused by our oversight. Please refer to Chapter 2.7 for details.
- Table 2: Include final weight and use DMI instead of ADFI.
au: Thank you for your suggestion. Following your recommendation, we are using DMI instead of ADFI. Please refer to Table 4 for more information. We have replaced ADFI with DMI in the article and marked it in red.
- Line 650-660: I recommend rewriting the conclusions so that they do not look like results.
au: Thank you for your suggestion. Based on your suggestions, make changes to the conclusion section. Please refer to Chapter 5 for details.
Kind regards
- Comments on the Quality of English Language
I noticed some grammatical errors that can be easily fixed.
au: Thank you for your suggestion. Grammatical changes have been applied throughout the article based on your suggestions. Please refer to the manuscript for details.
Again, many thanks for your valuable reviews.

Reviewer 2 Report
Comments and Suggestions for Authors
he research titled "Effects of Artemisia ordosica Crude Polysaccharide on Antioxidant and Immunity Capacity, Nutrient Digestibility, Rumen Fermentation, and Microbiota in Cashmere Goats" has caught my attention. The study aims to investigate the impact of Artemisia supplementation on various zootechnical and physiological parameters in cashmere goats. The research topic aligns with the journal's focus and is intriguing, considering the scarcity of articles on this particular goat breed. However, I believe that, in its current form, it has several shortcomings:
• The research title is quite lengthy, and I would recommend modifying it for improved clarity and readability. For instance, I suggest a title like "The Impact of Artemisia ordosica Polysaccharides on Cashmere Goats: Antioxidant and Immune Responses, Nutrient Digestion, Rumen Fermentation, and Microbiota." However, if the authors have a better title in mind, they are free to use it.
• I would suggest making slight adjustments to the abstract to include more numerical data. Additionally, I recommend avoiding overly long sentences in the abstract, as they may pose difficulties for some readers. Optimizing the abstract for readability is advisable.
• To enhance the research's attractiveness, I suggest refraining from using terms in the keywords that are already included in the article title. For instance, consider including terms such as "natural extracts" and "animal health."
• The introduction, in my opinion, requires a complete revision. I propose modifying it to provide a different perspective on the context. For instance, starting with a paragraph on the impact of adding specific plant compounds to animal health and productivity could be beneficial. You may want to refer to and cite articles such as "10.3390/ani13182967" and "10.3390/ani13050797." Afterward, proceed to describe the plant and its observed effects on other species.
• While the materials and methods section is well-written, I would like to request the addition of more specific details, such as the age and gender of the subjects used and the overall duration of the study. Additionally, it would be helpful to include information on the methods used for food sample collection. I also recommend expanding the section on statistical analysis.
The sample size and experimental design should be discussed in more detail. Providing information on the number of animals used, their characteristics, and the statistical methods employed would enhance the study's transparency and replicability.
Could you please clarify whether you conducted tests for normality and homogeneity on your data before proceeding with the statistical analysis? It's crucial to ensure that the assumptions underlying your chosen statistical methods are met. I recommend referring to the guidelines outlined in [proposed reference, e.g., 10.1080/1828051X.2020.1827990] for conducting such tests to maintain the rigor and reliability of your analysis.
The potential impact of aflatoxin levels in the feed, particularly in the corn used during the trial, on liver function and study outcomes is a valid concern. To address this issue and ensure the integrity of our study, we would like to confirm that rigorous quality control measures were implemented throughout the study. Specifically, the feed provided to the animals, including the corn, was regularly tested to ensure that aflatoxin levels remained well below established safety limits for animal consumption. This stringent monitoring was undertaken to mitigate any potential bias related to aflatoxin contamination, which could adversely affect liver health and consequently influence the study's results.
Please report a specific comment regarding the absence of this kind of bias of your study such as: "The diet provided in this study was carefully monitored to ensure that aflatoxin levels were well below the established safety limits for animal feed. This precautionary measure was taken to safeguard the animals' health and welfare. Aflatoxin contamination in animal feed can pose serious health risks, including impaired growth and liver damage (see, for example, 10.3390/toxins14070430). By maintaining feed quality within safe limits, we aimed to minimize any potential influence of aflatoxins on the study results."
• It's worth questioning why there are no data provided regarding the quality of meat and fur in the animals. Given the breed's significance in these productions (as emphasized in the introduction), I would have expected to see results related to these parameters.
It would be valuable to include an assessment of feed palatability in your study. Feed palatability is a critical factor influencing feed intake and, subsequently, animal performance. It can significantly affect the acceptance and consumption of specific feed components. Adding a section discussing feed palatability and citing relevant references in animal feeding practice would enhance the comprehensiveness of your study. Consider to cite: 10.1016/j.applanim.2020.105110
Please double-check the reference list to ensure that all references are included in the main text and vice versa.
Specific omments:
L141: Paragraph 2.3 is identical to 2.2, please remove it
Author Response
Response to Reviewer 2 Comments
Dear Reviewer,
Thank you very much for your precious comments and advice. Those comments are all valuable and very helpful for revising and improving our paper, as well as the important guiding significance to our researches. We have studied comments carefully and revised our manuscript based on them. The detailed responds to your review are attached below.
We would love to thank you for allowing us to resubmit a revised copy of the manuscript and we highly appreciate your time and consideration.
Kind regards,
Authors
- The research title is quite lengthy, and I would recommend modifying it for improved clarity and readability. For instance, I suggest a title like "The Impact of Artemisia ordosica Polysaccharides on Cashmere Goats: Antioxidant and Immune Responses, Nutrient Digestion, Rumen Fermentation, and Microbiota." However, if the authors have a better title in mind, they are free to use it.
au: Thank you for your suggestion. According to your suggestion, the article's title has been corrected. Please refer to lines 2-4 for details.
- I would suggest making slight adjustments to the abstract to include more numerical data. Additionally, I recommend avoiding overly long sentences in the abstract, as they may pose difficulties for some readers. Optimizing the abstract for readability is advisable.
au: Thank you for your suggestion. According to your suggestions, the abstract has been corrected. Please refer to lines 25-45 for details.
- To enhance the research's attractiveness, I suggest refraining from using terms in the keywords that are already included in the article title. For instance, consider including terms such as "natural extracts" and "animal health."
au: Thank you for your suggestion. According to your suggestion, we have added "natural extracts" and "animal health." Please refer to lines 46-47 for details.
4.The introduction, in my opinion, requires a complete revision. I propose modifying it to provide a different perspective on the context. For instance, starting with a paragraph on the impact of adding specific plant compounds to animal health and productivity could be beneficial. You may want to refer to and cite articles such as "10.3390/ani13182967" and "10.3390/ani13050797." Afterward, proceed to describe the plant and its observed effects on other species.
au: Thank you for your suggestion. Based on your suggestion, the paragraphs beginning with the effects of specific plant compounds on animal health and productivity have been added. We have cited an article ("10.3390/ani13182967"). Please refer to lines 64-69 for more details. Additionally, the references have been corrected. Please refer to lines 684-686 for more details.
5. While the materials and methods section is well-written, I would like to request the addition of more specific details, such as the age and gender of the subjects used and the overall duration of the study. Additionally, it would be helpful to include information on the methods used for food sample collection. I also recommend expanding the section on statistical analysis.
au: Thank you for your suggestions. According to your suggestion, the age and sex of the subjects have been added to chapter 2.2. Further details are provided in chapter 2.3 regarding the collection of feces and the determination of rumen fermentation parameters. Markings are in red font. Please refer to chapters 2.2 and 2.3 for more details.
- The sample size and experimental design should be discussed in more detail. Providing information on the number of animals used, their characteristics, and the statistical methods employed would enhance the study's transparency and replicability.
au: Thank you for your suggestions. The sample size and experimental design should be discussed in more detail in the Chapter 2.2 section based on your suggestion. Please refer to Chapter 2.2 for more information.
- Could you please clarify whether you conducted tests for normality and homogeneity on your data before proceeding with the statistical analysis? It's crucial to ensure that the assumptions underlying your chosen statistical methods are met. I recommend referring to the guidelines outlined in [proposed reference, e.g., 10.1080/1828051X.2020.1827990] for conducting such tests to maintain the rigor and reliability of your analysis.
au: According to your suggestion, all the data were tested for normal distribution. The data conformed to normal distribution was still analysed by Paired-Samples T Test. While the data not conformed to normal distribution, we re-analyzed using Kruskal Wallis test. Overall, our indicators, including ADG, DMI, F/G, Ca, CAT, IL-1β, IL-6, iNOS, NH3-N, Propionate, TVFA, Cyanobacteria, Verrucomicrobiota, Synergistota, and Actinobacteriota, were out of normal distribution. The above data were tested for accuracy using the rank sum test, with the corrected data highlighted in red. In contrast, all other indicators conform to a normal distribution with no value change. Consequential changes have been made in the tables marked in red text (Table 4, Table 5, Table 6, Table 7, Table 8, Table 9, and Table 10). Below is a table showing the normal distribution p-values.
Testing for normally distributed p-values.
|
Item |
P1-value |
P2-value |
P3-value |
|
Initial BW, kg |
0.717 |
0.936 |
0.769 |
|
ADG, g/d |
0.028 |
0.007 |
0.018 |
|
DMI, g/d |
0.001 |
0.715 |
0.764 |
|
F/G |
0.012 |
0.016 |
0.024 |
|
DM, % |
0.693 |
0.010 |
0.052 |
|
CP, % |
0.146 |
0.010 |
0.022 |
|
EE, % |
0.912 |
0.025 |
0.079 |
|
NDF, % |
0.073 |
0.023 |
0.148 |
|
ADF, % |
0.373 |
0.004 |
0.016 |
|
Ca, % |
0.009 |
0.262 |
0.265 |
|
P, % |
0.308 |
0.200 |
0.325 |
|
CAT, U/mL |
0.003 |
0.006 |
0.001 |
|
GSH-Px, U/mL |
0.762 |
0.010 |
0.036 |
|
MDA, nmol/mL |
0.955 |
0.936 |
0.819 |
|
T-SOD, U/mL |
0.130 |
0.010 |
0.039 |
|
T-AOC, U/mL |
0.261 |
0.873 |
0.173 |
|
TrxR, U/mL |
0.828 |
0.749 |
0.375 |
|
IL-1β, pg/mL |
0.006 |
0.199 |
0.262 |
|
IL-6, pg/mL |
0.006 |
0.003 |
0.005 |
|
ROS, pg/mL |
0.506 |
0.518 |
0.922 |
|
TNF-α, pg/mL |
0.950 |
0.518 |
0.687 |
|
NO, µmol/L |
0.255 |
0.004 |
0.002 |
|
iNOS, U/mL |
0.001 |
0.421 |
0.258 |
|
pH |
0.071 |
0.936 |
0.603 |
|
NH3-N, mg/100mL |
0.821 |
0.423 |
0.369 |
|
BCP, mg/100mL |
0.074 |
0.004 |
0.03 |
|
Protozoon, ×104/mL |
0.004 |
0.004 |
0.001 |
|
Acetate |
0.067 |
0.037 |
0.025 |
|
Propionate |
0.044 |
0.004 |
<0.001 |
|
Iso-butyrate |
0.439 |
0.007 |
0.015 |
|
Butyrate |
0.208 |
0.004 |
0.021 |
|
Iso-butyrate |
0.399 |
0.200 |
0.022 |
|
Valerate |
0.355 |
0.004 |
0.341 |
|
TVFA |
0.028 |
0.004 |
<0.001 |
|
A/P |
0.061 |
0.004 |
<0.001 |
|
Firmicutes |
0.662 |
0.150 |
0.224 |
|
Bacteroidota |
0.554 |
0.200 |
0.242 |
|
Patescibacteria |
0.200 |
1.000 |
0.754 |
|
Proteobacteria |
0.079 |
0.423 |
0.353 |
|
Cyanobacteria |
0.000 |
0.936 |
0.723 |
|
Verrucomicrobiota |
0.001 |
0.128 |
0.194 |
|
Spirochaetota |
0.054 |
0.629 |
0.894 |
|
Synergistota |
<0.001 |
0.470 |
0.259 |
|
Actinobacteriota |
0.044 |
0.631 |
0.849 |
|
other |
0.822 |
0.200 |
0.376 |
P1-value means whether the sample conforms to a normal distribution; when p1 > 0.05, it means it fits a normal distribution, which means using Paired-Samples T Test p-value (P3-value); when p1 < 0.05, it means it doesn't conform to a normal distribution, which means using Kruskal -Wallis test p-value (P2-value); P2-value = Kruskal -Wallis test p-value; P3-value = Paired-Samples T Test p-value.
- The potential impact of aflatoxin levels in the feed, particularly in the corn used during the trial, on liver function and study outcomes is a valid concern. To address this issue and ensure the integrity of our study, we would like to confirm that rigorous quality control measures were implemented throughout the study. Specifically, the feed provided to the animals, including the corn, was regularly tested to ensure that aflatoxin levels remained well below established safety limits for animal consumption. This stringent monitoring was undertaken to mitigate any potential bias related to aflatoxin contamination, which could adversely affect liver health and consequently influence the study's results. Please report a specific comment regarding the absence of this kind of bias of your study such as: "The diet provided in this study was carefully monitored to ensure that aflatoxin levels were well below the established safety limits for animal feed. This precautionary measure was taken to safeguard the animals' health and welfare. Aflatoxin contamination in animal feed can pose serious health risks, including impaired growth and liver damage (see, for example, 10.3390/toxins14070430). By maintaining feed quality within safe limits, we aimed to minimize any potential influence of aflatoxins on the study results."
au: Thanks for your question. I completely agree with you. We obtain corn and other ingredients in our diet from commercial sources that meet the requirements of national health standards. As a result, their aflatoxin levels are well below the established safety limits for animal feed. Following your suggestion, we have added specific comments to lines 131 and 132. Please refer to lines 131-132 for details.
- It's worth questioning why there are no data provided regarding the quality of meat and fur in the animals. Given the breed's significance in these productions (as emphasized in the introduction), I would have expected to see results related to these parameters.
au: Thank you for your question. Please accept my apologies for the questions you asked because we do not slaughter animals. Therefore, it is impossible to provide data on the quality of animals' meat and furs. Following your suggestion, relevant experiments could be conducted, and the precise results would need further discussion.
- It would be valuable to include an assessment of feed palatability in your study. Feed palatability is a critical factor influencing feed intake and, subsequently, animal performance. It can significantly affect the acceptance and consumption of specific feed components. Adding a section discussing feed palatability and citing relevant references in animal feeding practice would enhance the comprehensiveness of your study. Consider to cite: 10.1016/j.applanim.2020.105110
au: Thank you for your suggestion. Based on your recommendations, I have added the effect of palatability on growth performance. Please refer to lines 619-631 for details. References have been added, please refer to lines 838-845 for more information.
- Please double-check the reference list to ensure that all references are included in the main text and vice versa.
au: Thank you for your suggestion. Based on your suggestions, please ensure all references are included in the main text. Please see the manuscript for more information.
Specific omments:
L141: Paragraph 2.3 is identical to 2.2, please remove it
au: Thank you for your suggestion. The sequence number correction has been completed, and chapter 2.3 has been deleted. Please accept our apologies for any inconvenience caused by our oversight. Please consult the manuscript for details.
Again, many thanks for your valuable reviews.

Round 2
Reviewer 2 Report
Comments and Suggestions for Authors
Dear Authors,
I wanted to extend my heartfelt congratulations to you and your team for the outstanding job you've done in revising your paper. I am genuinely impressed by the way you have meticulously incorporated the suggested revisions. Your commitment to improving the article's quality is evident, and I must say that the final result is nothing short of exceptional. The transformation from the initial draft to the current version is remarkable and a testament to your dedication to excellence.